# Personalized Bone Reconstruction and Regeneration in the Treatment of Craniosynostosis

**Federica Tiberio** [1,†], **Ilaria Cacciotti** [2,†], **Paolo Frassanito** [3], **Giuseppina Nocca** [4,5], **Gianpiero Tamburrini** [3,6], **Alessandro Arcovito** [4,5,*] and **Wanda Lattanzi** [1,4,*]

1   Dipartimento Scienze della Vita e Sanità Pubblica, Università Cattolica del Sacro Cuore, 00168 Rome, Italy; federica.tiberio@unicatt.it
2   Dipartimento di Ingegneria, Università di Roma 'Niccolò Cusano', INSTM RU, 00166 Rome, Italy; ilaria.cacciotti@unicusano.it
3   UOC Neurochirurgia Infantile, Dipartimento di Scienze dell'Invecchiamento, Neurologiche, Ortopediche e della Testa-Collo, Fondazione Policlinico Universitario A. Gemelli-IRCCS, 00168 Rome, Italy; paolo.frassanito@policlinicogemelli.it (P.F.); gianpiero.tamburrini@unicatt.it (G.T.)
4   Fondazione Policlinico Universitario Agostino Gemelli IRCCS, 00168 Rome, Italy; Giuseppina.nocca@unicatt.it
5   Dipartimento di Scienze Biotecnologiche di Base, Cliniche Intensivologiche e Perioperatorie, Università Cattolica del Sacro Cuore, 00168 Rome, Italy
6   Dipartimento di Neuroscienze, Università Cattolica del Sacro Cuore, 00168 Rome, Italy
*   Correspondence: alessandro.arcovito@unicatt.it (A.A.); wanda.lattanzi@unicatt.it (W.L.); Tel.: +39-0630156706 (A.A.); +39-0630154464 (W.L.)
†   F.T. and I.C. contributed equally.

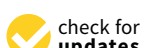

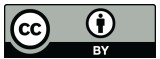

**Featured Application: This review outlines the extant techniques and innovative biotechnologies to implement the surgical treatment of craniosynostosis and related craniofacial deformities.**

**Abstract:** Craniosynostosis (CS) is the second most prevalent craniofacial congenital malformation due to the premature fusion of skull sutures. CS care requires surgical treatment of variable complexity, aimed at resolving functional and cosmetic defects resulting from the skull growth constrain. Despite significant innovation in the management of CS, morbidity and mortality still exist. Residual cranial defects represent a potential complication and needdedicated management to drive a targeted bone regeneration while modulating suture ossification. To this aim, existing techniques are rapidly evolving and include the implementation of novel biomaterials, 3D printing and additive manufacturing techniques, and advanced therapies based on tissue engineering. This review aims at providing an exhaustive and up-to-date overview of the strategies in use to correct these congenital defects, focusing on the technological advances in the fields of biomaterials and tissue engineering implemented in pediatric surgical skull reconstruction, i.e., biodegradable bone fixation systems, biomimetic scaffolds, drug delivery systems, and cell-based approaches.

**Keywords:** craniosynostosis; calvarial bone reconstruction; innovative biotechnologies; tissue engineering; personalized medicine; biomaterials; additive manufacturing; mesenchymal stromal cells; delivery systems

## 1. Introduction

Craniosynostosis (CS) as a group of disorders represents the second most prevalent congenital craniofacial malformation in humans (after cleft/lip palate), as it occurs in 1 out of about 2000 live births [1]. Like other craniofacial defects, craniosynostoses represent complex surgical challenges, as they require a multidisciplinary care and the need to cope with constitutive alteration of cell developmental programs due to underlying germline genetic mutations. This complexity often causes the inadequacy, the increased invasiveness and the related morbidity of existing reconstructive approaches [2]. For this reason,

extensive research efforts are in progress to design personalized strategies in CS treatments, implementing the development of novel biomaterials and production pipelines for bone tissue engineering.

In this context, the present review is aimed at providing an exhaustive overview of the current strategies in use to correct these congenital defects, focusing on the technological advances in the fields of biomaterials and tissue engineering implemented in pediatric surgical skull reconstruction. In particular, a lot of attention is dedicated to biodegradable bone fixation systems, biomimetic scaffolds, drug delivery systems, and cell-based approaches.

### 1.1. Craniosynostosis: A Heterogeneous Condition

Craniosynostosis (CS) is a congenital defect resulting from the premature fusion of one (simple or single-suture CS) or more (complex or multi-suture CS) skull sutures. This results in the abnormal growth of the skull, which is constrained perpendicularly to the fused suture and enhanced in a plane parallel to it, owing to the rapid brain growth underneath during the first years of postnatal life [3].

From a clinical point of view, the CS manifestations are heterogeneous, as it can be a feature in several distinct syndromes or it can occur as an isolated finding in nonsyndromic phenotypes [4,5]. Syndromic CS groups a wide variety of different syndromes with multisystem involvement, including CS in their phenotype. Table 1 lists all known syndromes featuring CS in their phenotype; the reported data include reference to the OMIM (Online Mendelian Inheritance in Man; www.omim.org (accessed on 2 July 2020)) ID, whenever available, or to the literature originally describing the genotype/phenotype. Nonsyndromic forms are usually single-suture CS, classified based on the involved suture site; these are listed in Table 2, with reference to the papers that have characterized the genetic etiology and/or the OMIM ID, whenever available.

The etiology of CS is also extremely heterogeneous. A genetic cause, being either a chromosomal structural aberration or a gene mutation, can be found in roughly 75–80% of syndromic forms [6]. An increasing number of disease-associated genes have been discovered during recent years and account for the allelic heterogeneity of CS (Table 1). Nonsyndromic CS (NCS) forms account for nearly three quarters of all craniosynostoses, being even more heterogeneous and unclear in their etiology [5,7]. Some NCS, at least those involving the sagittal suture, are believed to represent multifactorial disorders owing to the interplay between a significant genetic background and environmental risk factors [8]. Different genes have been recently associated with NCS, suggesting possible genotype/phenotype correlations between the mutated genes and the patterns of suture closure [5,7,9–19]. On this regard, it is worth noting that the presence of gene mutations may affect the neurodevelopmental prognosis in both syndromic and nonsyndromic CS patients [18,20].

The suture mesenchyme represents a unique skeletogenic stem cell niche in the developing skull, serving as a transient reservoir of mesenchymal stromal cells (MSC) and osteoprogenitors (see Figure 1) [21,22]. Regardless of the initial trigger underlying CS etiology, it is believed that in these disorders the calvarial suture stem cell niche undergoes an accelerated exhaustion, which ultimately drives the premature ossification of the suture mesenchyme [20,23]. Therefore, strategies to maintain and/or replenish the stem cell reservoir, either through direct cell transplantation, or through paracrine stimulation of endogenous cells, would be highly desirable.

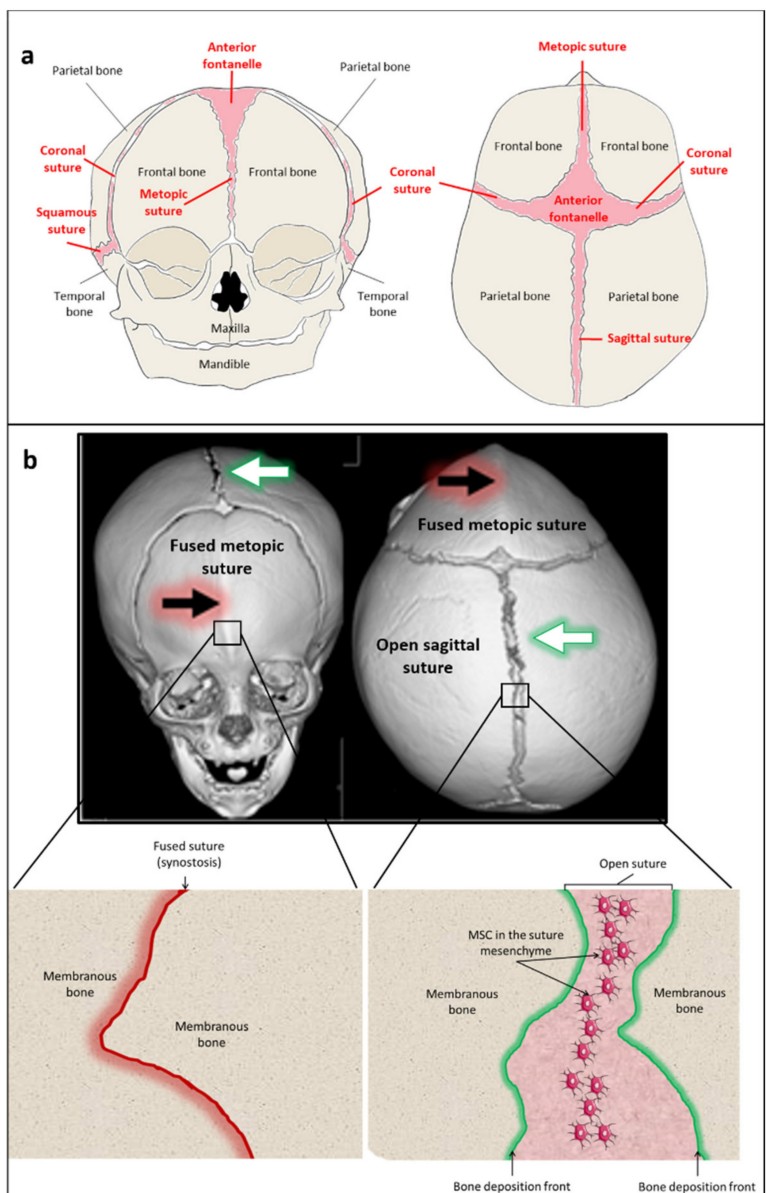

**Figure 1.** Skull sutures and craniosynostosis. The figure shows a structural and pathophysiological overview of craniosynostosis. (**a**) skull bones in the newborn skull, shown in frontal (**left**) and superior (**right**) views, are kept together by sutures (shown in pink); note that the frontal bone originates as two distinct bony pieces kept together by the metopic suture, which ossifies during early postnatal life. Fontanelles (i.e., palpable soft spots in the newborn skull) are found at crossroads between intersecting sutures and cause the skull to be elastic as needed to adapt dynamically to the rapid brain growth underneath. (**b**) 3D reconstruction of a patient's skull CT (computerized tomography) showing a synostosis of the metopic suture (the second most prevalent nonsyndromic craniosynostosis (CS), see text for details), in frontal (**left**) and superior (**right**) views; black arrows indicate the fused metopic suture white arrows indicate the open sagittal suture. The magnifications in the lower panels represent a diagram of the tissue architecture at the fused (**left**) and open (**right**) suture sites. Mesenchymal stromal cells (MSC) are viable within the open suture mesenchyme until the suture is ossified. Informed consent by both the parents of the patients was obtained prior to use the CT scan for assembling the figure.

CS heterogeneity can sometimes hamper an early and accurate diagnosis, which is mandatory to prompt a timely intervention. CS is treated by surgery for cranial vault remodeling (as detailed in the following paragraph), to relieve the constraint that may

cause increased intracranial pressure and correct the craniofacial deformity. Pediatric cranial vault surgery is inherently invasive and typically requires blood transfusion, which account for most of the public health burden of the disease [24,25].

Despite the lack of a clear consensus on the optimal timing for CS intervention, this is usually performed within the first year of life. An early surgical treatment allows reducing the invasiveness and complexity of the procedures and the risk of perioperative complications [26], significantly impacting on the risks of prolonged brain growth restraint, and yielding better cosmetic and functional outcomes [27–30].

Patients with complex and/or syndromic CS, especially in the presence of proven genetic causes, typically require multiple surgeries, with increased burden and consequently worse prognosis [31]. Indeed, a personalized surgical planning driven by the knowledge of the specific genetic background of the patient may dramatically improve the clinical and neurodevelopmental outcome [32].

**Table 1.** Syndromic craniosynostosis and associated genes.

| Syndrome(s) or Phenotype | OMIM ID and/or PubMed Reference | Involved Suture(s) | Gene Symbol |
| --- | --- | --- | --- |
| 3C SYNDROME-LIKE PHENOTYPE | 603527 | Sagittal | *DPH1* |
| 3MC SYNDROME 3 | 248340 | Lambdoid | *COLEC10* |
| 3MC1 SYNDROME 1 | 257920 | Lambdoid, coronal | *MASP1* |
| ACROCEPHALOPOLYDACTYLOUS DYSPLASIA | 200995 | - | - |
| ACROCEPHALOPOLYSYNDACTYLY TYPE III | 101120 | - | - |
| ACROMELIC FRONTONASAL DYSOSTOSIS | 603671 | Coronal, lambdoid | *ZSWIM6* |
| ALAGILLE SYNDROME 1 | 118450 | Coronal | *JAG1* |
| ANTLEY-BIXLER SYNDROME WITH GENITAL ANOMALIES AND DISORDERED STEROIDOGENESIS, ABS1 | 201750 | Coronal, lambdoid | *POR* |
| ANTLEY-BIXLER SYNDROME WITHOUT GENITAL ANOMALIES AND DISORDERED STEROIDOGENESIS, ABS2 | 207410 | Coronal, lambdoid | *FGFR2* |
| APERT SYNDROME | 101200 | Multisuture | *FGFR2* |
| ARTHROGRYPOSIS, CLEFT PALATE, CRANIOSYNOSTOSIS, AND IMPAIRED INTELLECTUAL DEVELOPMENT | 618265 | Coronal | *PPP3CA* |
| ATYPICAL MARFANOID SYNDROME WITH CRANIOSYNOSTOSIS | 616914 | Sagittal | *FBN1* |
| AU-KLINE SYNDROME | 616580 | Sagittal, metopic, multisuture | *HNRNPK* |
| AUROCEPHALOSYNDACTYLY | 109050 | - | - |
| B3GAT3-RELATED DISORDER | 606374 | Multisuture | *B3GAT3* |
| BALLER-GEROLD SYNDROME | 218600 | Multisuture | *RECQL4* |

**Table 1.** *Cont.*

| Syndrome(s) or Phenotype | OMIM ID and/or PubMed Reference | Involved Suture(s) | Gene Symbol |
|---|---|---|---|
| **BEARE-STEVENSON CUTIS GYRATA SYNDROME** | 123790 | Multisuture | *FGFR2* |
| **BENT BONE DYSPLASIA SYNDROME** | 614592 | Coronal | *FGFR2* |
| **BOHRING-OPITZ SYNDROME** | 605039 | Metopic | *ASXL1* |
| **BRACHYCEPHALY, DEAFNESS, CATARACT, MICROSTOMIA, AND MENTAL RETARDATION** | % 601353 | Coronal, lambdoid | - |
| **BRACHYDACTYLY, TYPE C** | 113100 | Variable | *GDF5* |
| **BRAIN MALFORMATIONS WITH OR WITHOUT URINARY TRACT DEFECTS** | 613735 | Sagittal, lambdoid | *NFIA* |
| **CARDIO-FACIO-CUTANEOUS SYNDROME** | 115150 | Sagittal | *BRAF* |
| **CARNEVALE SYNDROME** | 265050 | Coronal | *COLEC11* |
| **CARPENTER SYNDROME 1** | 201000 | Sagittal, lambdoid, coronal | *RAB23* |
| **CARPENTER SYNDROME 2** | 614976 | Multisuture | *MEGF8* |
| **CEBALID SYNDROME** | 618774 | Variable | *MN1* |
| **CEREBROOCULONASAL SYNDROME** | % 605627 | - | - |
| **CHAR SYNDROME** | 169100 | Lambdoid | *TFAP2B* |
| **CHERUBISM** | 118400 | Sagittal, coronal | *SH3BP2* |
| **CHONDRODYSPLASIA WITH JOINT DISLOCATIONS, GRAPP TYPE** | 614078 | Coronal | *IMPAD1* |
| **CHROMOSOME 10Q26 DELETION SYNDROME** | 609625 | Metopic | *10q26* [a] |
| **CHROMOSOME 19P13.13 DELETION SYNDROME** | 613638 | Coronal, lambdoid, parieto-temporal | *(9p13.2–p13.13)* [a] |
| **CHROMOSOME 5P13 DUPLICATION SYNDROME** | 613174 | - | 5p13 [a] |
| **CHROMOSOME 9P DELETION SYNDROME** | 158170 | Metopic | 9p [a] |
| **COFFIN-SIRIS SYNDROME 7** | 618027 | Sagittal, metopic | *DPF2* |
| **COLE-CARPENTER SYNDROME 1** | 112240 | Multisuture | *P4HB* |
| **COLE-CARPENTER SYNDROME 2** | 616294 | Sagittal | *SEC24D* |
| **CONGENITAL DISORDER OF GLYCOSYLATION, TYPE IIn** | 616721 | Coronal, lambdoid | *SLC39A8* |
| **CONTRACTURES, PTERYGIA, AND SPONDYLOCARPOTARSAL FUSION SYNDROME 1A** | 178110 | - | *MYH3* |
| **CORNELIA DE LANGE SYNDROME 1** | 122470 | - | *NIPBL* |

**Table 1.** *Cont.*

| Syndrome(s) or Phenotype | OMIM ID and/or PubMed Reference | Involved Suture(s) | Gene Symbol |
|---|---|---|---|
| CRANIOECTODERMAL DYSPLASIA 1 | 218330 | Sagittal | *IFT122* |
| CRANIOECTODERMAL DYSPLASIA 2 SYNDROME | 613610 | Sagittal | *WDR35* |
| CRANIOECTODERMAL DYSPLASIA 3 | 614099 | Sagittal | *IFT43* |
| CRANIOECTODERMAL DYSPLASIA 4 SYNDROME (FRONTAL BOSSING) | 614378 | Sagittal | *WDR19* |
| CRANIOFACIAL DYSMORPHISM, SKELETAL ANOMALIES, AND MENTAL RETARDATION SYNDROME | 213980 | Multisuture | *TMCO1* |
| CRANIOFACIAL DYSSYNOSTOSIS WITH SHORT STATURE | [33] (218350) | Sagittal, lambdoid, coronal | *SOX6* |
| CRANIOFACIAL-SKELETAL-DERMATOLOGIC DYSPLASIA | 101600 | Multisuture | *FGFR2* |
| CRANIOFACIOCUTANEOUS SYNDROME | 164757 | Sagittal, lambdoid | *BRAF* |
| CRANIOFRONTONASAL SYNDROME | 304110 | Coronal | *EFNB1* |
| CRANIOMICROMELIC SYNDROME | 602558 | Coronal | - |
| CRANIORHINY | 123050 | - | - |
| CRANIOSTENOSIS, SAGITTAL, WITH CONGENITAL HEART DISEASE, MENTAL DEFICIENCY, AND MANDIBULAR ANKYLOSIS | 218450 | Sagittal | - |
| CRANIOSYNOSTOSIS 1 | 123100 | Coronal, sagittal | *TWIST1* |
| CRANIOSYNOSTOSIS 2 (BOSTON-TYPE) | 604757 | Multisuture | *MSX2* |
| CRANIOSYNOSTOSIS 3 | 615314 | Coronal, sagittal | *TCF12* |
| CRANIOSYNOSTOSIS 4 | 600775 | Sagittal, lambdoid, coronal, metopic or multisuture | *ERF* |
| CRANIOSYNOSTOSIS 5 | 615529 | Sagittal | *ALX4* |
| CRANIOSYNOSTOSIS 6 | 616602 | Coronal | *ZIC1* |
| CRANIOSYNOSTOSIS 7 | 617439 | Sagittal, metopic | *SMAD6* |
| CRANIOSYNOSTOSIS AND DENTAL ANOMALIES | 614188 | Metopic, coronal, sagittal and/or lambdoid | *IL11RA* |
| CRANIOSYNOSTOSIS SYNDROME, AUTOSOMAL RECESSIVE | 606529 | Variable | - |
| CRANIOSYNOSTOSIS WITH ANOMALIES OF THE CRANIAL BASE AND DIGITS | 218530 | Coronal, lambdoid | - |

**Table 1.** *Cont.*

| Syndrome(s) or Phenotype | OMIM ID and/or PubMed Reference | Involved Suture(s) | Gene Symbol |
| --- | --- | --- | --- |
| **CRANIOSYNOSTOSIS WITH ECTOPIA LENTIS** | 603595 | Coronal or multisuture | *ADAMTSL4* |
| **CRANIOSYNOSTOSIS WITH FIBULAR APLASIA** | 281550 | Coronal, sagittal | - |
| **CRANIOSYNOSTOSIS WITH OCULAR ABNORMALITIES AND HALLUCAL DEFECTS** | 608279 | Coronal, sagittal | - |
| **CRANIOSYNOSTOSIS WITH RADIOHUMERAL FUSIONS AND OTHER SKELETAL AND CRANIOFACIAL ANOMALIES** | 614416 | Coronal, lambdoid, multisuture | *CYP26B1* |
| **CRANIOSYNOSTOSIS, ADELAIDE TYPE; CRSA** | % 600593 | - | *MSX1, FGFR3* (4p16 [a]) |
| **CRANIOSYNOSTOSIS, CALCIFICATION OF BASAL GANGLIA, AND FACIAL DYSMORPHISM** | 608432 | - | - |
| **CRANIOSYNOSTOSIS-MENTAL RETARDATION SYNDROME OF LIN AND GETTIG** | 218649 | Sagittal, lambdoid, metopic | - |
| **CRANIOSYNOSTOSIS-MENTAL RETARDATION-CLEFTING SYNDROME** | 218650 | - | - |
| **CRANIOTELENCEPHALIC DYSPLASIA** | 218670 | - | - |
| **CROUZON SYNDROME** | 123500 | Multisuture | *FGFR2* |
| **CROUZON WITH ACANTHOSIS NIGRICANS SYNDROME** | 612247 | Coronal, multisuture | *FGFR3* |
| **CURRY-JONES SYNDROME** | 601707 | Coronal | *SMOH* |
| **DEVELOPMENTAL DELAY WITH SHORT STATURE, DYSMORPHIC FEATURES, AND SPARSE HAIR** | 616901 | Sagittal, metopic | *DPH1* |
| **DEVELOPMENTAL DELAY WITH VARIABLE INTELLECTUAL IMPAIRMENT AND BEHAVIORAL ABNORMALITIES** | 618430 | Coronal, metopic, multisuture | *TCF20* |
| **DIABETES MELLITUS, NEONATAL, WITH CONGENITAL HYPOTHYROIDISM** | 610199 | Sagittal | *GLIS3* |
| **DISTAL CHROMOSOME 7Q11.23 DELETION SYNDROME** | 613729 | - | 7q11.23 [a] |
| **DUBOWITZ SYNDROME** | % 223370 | - | - |
| **ELLIS VAN CREVELD SYNDROME** | 225500 | Sagittal | *EVC* |
| **ENDOSTEAL HYPEROSTOSIS** | 144750 | Multisuture | *LRP5* |

**Table 1.** *Cont.*

| Syndrome(s) or Phenotype | OMIM ID and/or PubMed Reference | Involved Suture(s) | Gene Symbol |
|---|---|---|---|
| **FONTAINE PROGEROID SYNDROME** | 612289 | Coronal | *SLC25A24* |
| **FRANK-TER HAAR SYNDROME** | 249420 | Sagittal | *SH3PXD2B* |
| **FRONTONASAL DYSPLASIA 1** | 136760 | Coronal | *ALX3* |
| **FRONTONASAL DYSPLASIA 2** | 613451 | Coronal | *ALX4* |
| **FRONTONASAL DYSPLASIA VARIANT** | [34] | Sagittal, multisuture | *SIX2* |
| **FRONTOOCULAR SYNDROME** | 605321 | Coronal, metopic | - |
| **FRYNS MICROPHTHALMIA SYNDROME** | 600776 | - | - |
| **GABRIELE-DE VRIES SYNDROME** | 617557 | - | *YY1* |
| **GOLDBERG-SHPRINTZEN MEGACOLON SYNDROME** | 609460 | Variable | *KIAA1279* |
| **GOMEZ-LOPEZ-HERNANDEZ SYNDROME** | %601583 | Lambdoid | - |
| **GRACILE BONE DYSPLASIA** | 602361 | Skull base, multisuture | *FAM111A* |
| **GREIG CEPHALOPOLYSYNDACTYLY SYNDROME** | 175700 | Sagittal, metopic | *GLI3* |
| **GROWTH RETARDATION, DEVELOPMENTAL DELAY, FACIAL DYSMORPHISM** | 612938 | Coronal | *FTO* |
| **HAMAMY SYNDROME** | 611174 | Metopic | *IRX5* |
| **HARTSFIELD SYNDROME** | 615465 | - | *FGFR1* |
| **HENNEKAM LYMPHANGIECTASIA-LYMPHEDEMA SYNDROME 1** | 235510 | Coronal | *CCBE1* |
| **HOLOPROSENCEPHALY, SEMILOBAR, WITH CRANIOSYNOSTOSIS** | 601370 | Coronal, lamboid | - |
| **HUNTER-MCALPINE CRANIOSYNOSTOSIS SYNDROME** | 601379 | - | - |
| **HYPER-IGE RECURRENT INFECTION SYNDROME** | 147060 | Multisuture | *STAT3* |
| **HYPER-IGE RECURRENT INFECTION SYNDROME 4, AUTOSOMAL RECESSIVE** | 618523 | Multisuture | *IL6ST* |
| **HYPERTELORISM, TEEBI TYPE** | 145420 | Sagittal, coronal | *SPECC1L* |
| **HYPOPHOSPHATASIA, INFANTILE** | 241500 | - | *ALPL* |
| **HYPOPHOSPHATEMIC RICKETS, X-LINKED DOMINANT** | 307800 | Sagittal | *PHEX* |

Table 1. *Cont.*

| Syndrome(s) or Phenotype | OMIM ID and/or PubMed Reference | Involved Suture(s) | Gene Symbol |
|---|---|---|---|
| **IMAGE SYNDROME** | 614732 | Sagittal, metopic | *CDKN1C* |
| **IMMUNOSKELETAL DYSPLASIA WITH NEURODEVELOPMENTAL ABNORMALITIES** | 617425 | Multisuture | *EXTL3* |
| **JACKSON-WEISS SYNDROME** | 123150 | Multisuture | *FGFR1* |
| **JACKSON-WEISS SYNDROME** | 123150 | Multisuture | *FGFR2* |
| **JOUBERT SYNDROME 2** | 608091 | Sagittal | *TMEM216* |
| **KABUKI SYNDROME** | 147920 | Multisuture | *KMT2D* |
| **KAKUBI SYNDROME/AU-KLINE SYNDROME** | 600712 | Metopic | *HNRNPK* |
| **KLEEBLATTSCHAEDEL** | 148800 | - | - |
| **LACRIMOAURICULODENTODIGITAL (LEVY-HOLLISTER) SYNDROME** | 149730 | Sagittal | *FGF10* |
| **LIN-GETTING SYNDROME-LIKE CSO/GENITOPATELLAR SYNDROME/SAY BARBER BIESECKER YOUNG SIMPSON SYNDROME** | 605880 | Sagittal | *KAT6B* |
| **LOEYS-DIETZ SYNDROME 1** | 609192 | Multisuture | *TGFBR1* |
| **LOEYS-DIETZ SYNDROME 2** | 610168 | Multisuture | *TGFBR2* |
| **LOEYS-DIETZ SYNDROME 3** | 613795 | Variable | *SMAD3* |
| **LOEYS-DIETZ SYNDROME 4** | 614816 | Sagittal, metopic, multisuture | *TGFB2* |
| **MEIER-GORLIN SYNDROME (ATYPICAL)** | 224690 | Coronal | *CDC45* |
| **MEIER-GORLIN SYNDROME 1** | 224690 | Variable | *ORC1* |
| **MEIER-GORLIN SYNDROME 7** | 617063 | Coronal, multisure | *CDC45L* |
| **MENTAL RETARDATION, AR 41** | 615637 | Sagittal | *KPTN* |
| **MENTAL RETARDATION, AUTOSOMAL DOMINANT 32** | 616268 | Coronal | *KAT6A* |
| **MENTAL RETARDATION, AUTOSOMAL DOMINANT 57** | 618050 | Coronal, sagittal, metopic, multisuture | *TLK2* |
| **MENTAL RETARDATION, X-LINKED, SYNDROMIC, TURNER TYPE** | 309590 | Metopic | *HUWE1* |
| **METAPHYSEAL ACROSCYPHODYSPLASIA** | % 250215 | - | - |
| **METOPIC/PANSYNOSTOSIS (DUPLICATION)** | [5] | Metopic, multisuture | *RUNX2* |
| **MICROCEPHALY 1, PRIMARY, AUTOSOMAL RECESSIVE** | 251200 | Variable | *MCPH1* |

**Table 1.** *Cont.*

| Syndrome(s) or Phenotype | OMIM ID and/or PubMed Reference | Involved Suture(s) | Gene Symbol |
|---|---|---|---|
| MICROCEPHALY WITH CHEMOTACTIC DEFECT AND TRANSIENT HYPOGAMMA-GLOBULINEMIA | 251240 | - | - |
| MICROCEPHALY, SHORT STATURE, AND POLYMICROGYRIA WITH OR WITHOUT SEIZURES | 614833 | - | *RTTN* |
| MICROCEPHALY-MICROMELIA SYNDROME | 251230 | - | *DONSON* |
| MICROPHTHALMIA, SYNDROMIC 6 | 607932 | Lambdoid | *BMP4* |
| MOSAIC VARIEGATED ANEUPLOIDY SYNDROME 2 | 614114 | Sagittal | *CEP57* |
| MOWAT-WILSON SYNDROME | 235730 | Coronal | *ZEB2* |
| MUCOLIPIDOSIS II | 252500 | Multisuture | *GNPTAB* |
| MUENKE NONSYNDROMIC CORONAL CRANIOSYNOSTOSIS | 602849 | Coronal | *FGFR3* |
| MUENKE SYNDROME | 602849 | Coronal, multisuture | *FGFR3* |
| MULTIPLE SYNOSTOSES SYNDROME 3 | 612961 | Variable | *FGF9* |
| NABLUS MASK-LIKE FACIAL SYNDROME | 608156 | - | 8q22.1 [a] |
| NAIL-PATELLA SYNDROME | 161200 | Coronal | *LMX1B* |
| NOONAN SYNDROME | 176876 | Sagittal | *PTPN11* |
| NOONAN SYNDROME 3 | 609942 | Sagittal | *KRAS* |
| NOONAN SYNDROME-LIKE DISORDER WITH LOOSE ANAGEN HAIR 2 | 617506 | Sagittal, coronal | *PPP1CB* |
| OBESITY, HYPERPHAGIA, AND DEVELOPMENTAL DELAY | 613886 | Coronal | *NTRK2* |
| OPITZ C SYNDROME | 211750 | Metopic | *CD96* |
| OPITZ GBBB SYNDROME TYPE II | 145410 | Metopic, sagittal | *SPECCL1* |
| OPITZ-KAVEGGIA SYNDROME | 305450 | Lambdoid, sagittal, multisuture | *MED12* |
| OSTEOGENESIS IMPERFECTA, TYPE VII | 610682 | Lambdoid, coronal, multisuture | *CRTAP* |
| OSTEOGLOPHONIC DYSPLASIA | 166250 | Coronal, multisuture | *FGFR1* |
| OSTEOPETROSIS, AUTOSOMAL RECESSIVE 5 | 259720 | Sagitall, coronal, multisuture | *OSTM1* |
| OTOPALATODIGITAL SPECTRUM DISORDERS WITH CS | 300017 | Skull base, Multisuture | *FLNA* |

**Table 1.** *Cont.*

| Syndrome(s) or Phenotype | OMIM ID and/or PubMed Reference | Involved Suture(s) | Gene Symbol |
|---|---|---|---|
| PFEIFFER SYNDROME | 101600 | Multisuture | *FGFR1* |
| PFEIFFER SYNDROME | 101600 | Multisuture | *FGFR2* |
| PHOSPHOSERINE AMINOTRANSFERASE DEFICIENCY | 610992 | Multisuture | *PSAT1* |
| PSEUDOHYPOPARATHYROIDISM TYPE 1 | 103580 | Coronal, metopic and sagittal | *GNAS* |
| PYCNODYSOSTOSIS (ATYPICAL) | 265800 | Coronal | *CTSK* |
| RAINE SYNDROME | 259775 | Coronal or multisuture | *FAM20C* |
| RETINITIS PIGMENTOSA WITH OR WITHOUT SKELETAL ANOMALIES | 250410 | - | *CWC27* |
| ROBERTS SYNDROME | 268300 | Multisuture | *ESCO2* |
| ROBINOW-SORAUF SYNDROME | 180750 | Coronal | *TWIST1* |
| SAETHRE-CHOTZEN SYNDROME | 101400 | Coronal, multisuture | *FGFR2* |
| SAETHRE-CHOTZEN SYNDROME WITH OR WITHOUT EYELID ANOMALIES | 101400 | Coronal, multisuture | *TWIST1* |
| SC PHOCOMELIA SYNDROME | 269000 | - | *ESCO2* |
| SCAPHOCEPHALY, MAXILLARY RETRUSION, AND MENTAL RETARDATION SYNDROME | 609579 | Sagittal | *FGFR2* |
| SEVERE COMBINED IMMUNODEFICIENCY WITH MICROCEPHALY, GROWTH RETARDATION, AND SENSITIVITY TO IONIZING RADIATION | 611291 | Multisuture | *NHEJ1* |
| SHORT-RIB THORACIC DYSPLASIA 13 WITH OR WITHOUT POLYDACTYLY | 616300 | Coronal | *CEP120* |
| SHORT-RIB THORACIC DYSPLASIA 6 WITH OR WITHOUT POLYDACTYLY | 263520 | Coronal | *NEK1* |
| SHORT-RIB THORACIC DYSPLASIA 9 WITH OR WITHOUT POLYDACTYLY SYNDROME | 266920 | Sagittal | *IFT140* |
| SHPRINTZEN-GOLDBERG CRANIOSYNOSTOSIS SYNDROME | 182212 | Coronal, sagittal or lambdoid | *SKI* |
| SPONDYLOEPIMETAPHYSEAL DYSPLASIA, FADEN-ALKURAYA TYPE | 616723 | Coronal | *RSPRY1* |

**Table 1.** *Cont.*

| Syndrome(s) or Phenotype | OMIM ID and/or PubMed Reference | Involved Suture(s) | Gene Symbol |
|---|---|---|---|
| SPONDYLOEPIPHYSEAL DYSPLASIA WITH CORONAL CRANIOSYNOSTOSIS, CATARACTS, CLEFT PALATE, AND MENTAL RETARDATION | 602611 | Coronal | - |
| STRUCTURAL BRAIN ANOMALIES WITH IMPAIRED INTELLECTUAL DEVELOPMENT AND CRANIOSYNOSTOSIS | 618736 | Coronal | *ZIC1* |
| SUMMITT SYNDROME | 272350 | - | - |
| SWEENEY-COX SYNDROME | 617746 | Variable | *TWIST1* |
| SYNDACTYLY, TYPE 1, WITH OR WITHOUT CRANIOSYNOSTOSIS | 185900 | Sagittal | *IHH* |
| TETRASOMY 15Q26 | 614846 | Metopic, coronal, multisuture | 15q26-qter [a] |
| THANATOPHORIC DYSPLASIA, TYPE I | 187600 | - | *FGFR3* |
| TOE SYNDACTYLY, TELECANTHUS, AND ANOGENITAL AND RENAL MALFORMATIONS | 300707 | - | *FAM58A* |
| TREACHER COLLINS SYNDROME 1 | 154500 | Multisuture | *TCOF1* |
| TRICHOTHIODYSTROPHY 6, NONPHOTOSENSITIVE | 616943 | Coronal | *GTF2E2* |
| TRIGONOCEPHALY 1 | 190440 | Multisuture | *FGFR1* |
| TRIGONOCEPHALY 2 | 614485 | Metopic | *FREM1* |
| TRIGONOCEPHALY WITH SHORT STATURE AND DEVELOPMENTAL DELAY | 314320 | Metopic | - |
| UNDEFINITED/UNCLEAR | 600921 | Sagittal | *FGF9* |
| UNDEFINITED/UNCLEAR | 610966 | Multisuture | *FTO* |
| UNDEFINITED/UNCLEAR | 147370 | Sagittal, coronal | *IGF1R* |
| UNDEFINITED/UNCLEAR | 600727 | Metopic | *NFIA* |
| UNDEFINITED/UNCLEAR | 611909 | - | *FNDC3B* |
| VAN DEN ENDE-GUPTA SYNDROME | 600920 | Multisuture | *SCARF2* |
| WEISS-KRUSZKA SYNDROME | 618619 | Metopic | *ZNF462* |
| WILLIAMS-BEUREN SYNDROME | 194050 | Sagittal | 7q11.23 [a] |
| ZTTK SYNDROME | 617140 | Metopic, sagittal, multisuture | *SON* |

[a] Cytogenetic location.

**Table 2.** Nonsyndromic Craniosynostosis and associated genes.

| Syndrome(s) or Phenotype | OMIM ID/ Reference | Involved Suture(s) | Gene Symbol |
|---|---|---|---|
| Craniosynostosis, nonsyndromic unicoronal | [35] | Coronal | *FGFR2* |
| Nonsyndromic craniosynostosis | [36] | - | *SNAI1* [a] |
| Nonsyndromic craniosynostosis | [37] | - | *PTH2R* (*intron break*) |
| Nonsyndromic craniosynostosis | [17] | Coronal | *EFNB1* [a] |
| Nonsyndromic craniosynostosis | [17] | Sagittal | *ALX4* [a] |
| Nonsyndromic craniosynostosis | [17] | Sagittal, coronal | *TWIST1* [a] (*c.435G>C*) |
| Nonsyndromic craniosynostosis | [17] | Coronal | *TWIST1* [a] (*c.421G>C*) |
| Nonsyndromic craniosynostosis | [17] | Sagittal | *ADCK1* [a] |
| Nonsyndromic craniosynostosis | [17] | Sagittal | *ALPL* [a] |
| Nonsyndromic craniosynostosis | [17] | Sagittal | *BMPER* [a] |
| Nonsyndromic craniosynostosis | [17] | Sagittal, coronal | *FREM1* [a] |
| Nonsyndromic craniosynostosis | [17] | Sagittal, coronal | *FREM1* [a] |
| Nonsyndromic craniosynostosis | [17] | Coronal | *JAG1* [a] |
| Nonsyndromic craniosynostosis | [17] | Coronal | *NELL1* [a] |
| Nonsyndromic craniosynostosis | [17] | Sagittal | *NOTCH 1* [a] |
| Nonsyndromic craniosynostosis | [17] | Sagittal | *NOTCH2* [a] |
| Nonsyndromic craniosynostosis | [17] | Sagittal | *PDILT* [a] |
| Nonsyndromic craniosynostosis | [17] | Sagittal | *REQL4* [a] |
| Nonsyndromic craniosynostosis | [17] | Coronal | *SHC4* [a] |
| Nonsyndromic craniosynostosis | [17] | Sagittal | *TGFBR2* [a] |
| Nonsyndromic craniosynostosis | [38] | Metopic | *RUNX2* [b] |
| Nonsyndromic midline craniosynostosis | [18] | Metopic | *ARAP3* |

**Table 2.** *Cont.*

| Syndrome(s) or Phenotype | OMIM ID/ Reference | Involved Suture(s) | Gene Symbol |
|---|---|---|---|
| Nonsyndromic midline craniosynostosis | [18] | Sagittal | AXIN1 |
| Nonsyndromic midline craniosynostosis | [18,39] (* 112261) | Sagittal, metopic | BMP2 [b] |
| Nonsyndromic midline craniosynostosis | [18] | Sagittal | DVL3 |
| Nonsyndromic midline craniosynostosis | [18] | Sagittal | MESP1 |
| Nonsyndromic midline craniosynostosis | [18] | Sagittal | NPHP4 |
| Nonsyndromic midline craniosynostosis | [18] | Metopic | PSMC2 |
| Nonsyndromic midline craniosynostosis | [18] | Metopic | PSMC5 |
| Nonsyndromic midline craniosynostosis | [18] | Sagittal | RASAL2 |
| Nonsyndromic midline craniosynostosis | [18,39] (* 602931) | Sagittal and metopic | SMAD6 |
| Nonsyndromic midline craniosynostosis | [18] | Metopic | SMURF1 |
| Nonsyndromic midline craniosynostosis | [18] (* 602465) | Sagittal | SPRY1 |
| Nonsyndromic midline craniosynostosis | [18] (* 607984) | Sagittal | SPRY4 |
| Nonsyndromic midline craniosynostosis | [10] | Sagittal, metopic | BBS9 [a] |
| Nonsyndromic unicoronal synostosis | [40] | Coronal | EFNA4 |
| Sagittal nonsyndromic craniosynostosis | [19] | Sagittal, lambdoid | FGFR2 |
| Sagittal nonsyndromic craniosynostosis | [41] | Sagittal, metopic | FGFR3 |
| Sagittal nonsyndromic craniosynostosis | [42] (* 147370) | Sagittal, metopic, coronal | IGF1R |

**Table 2.** *Cont.*

| Syndrome(s) or Phenotype | OMIM ID/ Reference | Involved Suture(s) | Gene Symbol |
|---|---|---|---|
| Sagittal nonsyndromic craniosynostosis | [43] | Sagittal, multisuture | *LRIT3* |
| Sagittal nonsyndromic craniosynostosis | [19] | Sagittal, coronal | *TWIST1 (c.563 > T)* |

[a] Variants predicted to be pathogenetic in nonsyndromic craniosynostosis patients; [b] susceptibility to CS. * an asterisk preceeding the OMIM ID indicates a gene entry (rather than a phenotype) in the catalogue.

### 1.2. Overview of Surgical Approaches and Skull Reconstructive Techniques

Aims of the surgical treatment of CS are (i) to correct the functional and cosmetic anomalies, (ii) to restore the normal spatial relationships between the skull and the contained cerebral and vascular structures, (iii) to reorient the abnormal vectors of cranial growth, and (iv) to correct the possibly associated alterations in cerebrospinal fluid (CSF) dynamics and venous circulation.

In syndromic craniosynostosis, timing and choice of the surgical procedure result from the complex interpretation of anatomic and functional anomalies and of their variable inter-reaction in different patients at different stages of the disease. When the cranial vault is taken into consideration, posterior cranial vault expansion is indicated when the posterior fossa is volumetrically reduced up to determine a secondary Chiari malformation. Conversely, whenever ocular and respiratory functions are sufficiently preserved, fronto-orbital and maxillary advancement are postponed as close as possible to the end of maxillo-facial growth, namely around the seventh year of age [44].

In simple craniosynostosis, timing of the surgical treatment is more standardized, mainly depending on the age at diagnosis and the type of craniosynostosis. In the case of sagittal craniosynostosis, early surgery may be considered if the diagnosis is made in the first four months of life. Main advantages of early surgery in this subset of children are that correction may be achieved by means of minimally invasive cranial expansion techniques and that the relief on intracranial structures is anticipated with benefit on the early phases of brain development [45].

In children with craniosynostosis involving the coronal ring and the orbits (i.e., metopic and coronal craniosynostosis), open cranial vault remodeling is the preferred surgical technique. Surgery, in this case, is performed starting from the fifth–sixth month of life, an age warranting a more reliable stabilization of the reconstructed cranial vault, thanks to the higher consistency of the bone. In this context, the introduction of bioabsorbable plates and screws has reduced the complications related to the use of titanium devices to fix the bone structures [46], as discussed in the next paragraph.

A more recent concept in the surgical treatment of craniosynostosis is represented by distraction osteogenesis (DO): this is a method of generating new bone following a corticotomy or an osteotomy and gradual distraction. The method is based on the tension-stress principle proposed by Ilizarov [47,48]. The gradual bone distraction creates mechanical stimulation which induces biological responses, including differentiation of pluripotential cells, angiogenesis, osteogenesis, and bone mineralization, finally resulting in bone regeneration. DO techniques claim various advantages over the conventional cranial vault remodeling techniques, including a shorter surgical time, less bleeding, and good blood supply to the cranial bone because of limited dissection of the *dura mater*, and safe and large expansion of skull, because of simultaneous soft tissue expansion. It can be accomplished either by means of springs or distractors [49].

One of the main disadvantages of distraction methods is the difficulty to control the vectors of distraction, with a consequent increased risk of mechanical complications, such as unwarranted less expansion or overexpansion an undesirable cosmetic outcome, as

well as dislodgement of the device [49,50]. A second relevant drawback of springs and distractors is the need of a second surgery to remove the implanted device.

## 2. Biomaterials and Tissue Engineering Approaches

### 2.1. Biodegradable Rigid Fixation Systems

For the rigid fixation of the fragments during the craniosynostosis surgery, initially wire sutures were used, later titanium miniplates [46]. Then, biodegradable osteosynthesis materials were introduced [51], in order to avoid a second operation, needed to remove the metallic plates, commonly 3–6 months postsurgery, due to their migration in a centripetal direction [52] as a consequence of the appositional growth of cranial bone.

Table 3 lists the most commonly used materials used in fixation devices that are commercially available, with their composition and post-operative infection rates [53].

Among them, LactoSorb® is a widely used synthetic resorbable biomaterial for pediatric tissue repair that fully resorbs within 6–12 months. Diverse groups reported the outcome of bone reconstruction surgeries in CS patients using LactoSorb®; the cumulative frequency of post-operative infections in the different studies ranges from 0 to 3% (Table 3), in over 2500 patients overall, followed up until 12 months or more [54–62].

Other groups tested alternative PLLA-based biodegradable fixation systems for craniofacial implants, namely, the Biosorb® device (Table 3), tested in a single study on 161 patients [63], and the MacroPore®, evaluated in two independent studies including overall 86 patients [55,64].

Further, Inion® CPS baby, which consists in a blend of trimethylene carbonate, polylactic acid, and L-Lactide, D,L-Lactide, Polyglycolide (PDLLA), was used by different groups proving very low disadvantages in terms of post-operative infection risk (Table 3) [65–68].

Despite the reported advantages of efficient resorption and low post-operative infection rates, these biodegradable osteosynthesis materials are less stable and more challenging in their usage [69]. Among the major drawbacks, these devices present a complicated handling and time-consuming thread cutting with respect to the titanium plates. Furthermore, screw fixation of resorbable plates leads to stable results only when the screws are accurately applied in an orthograde direction. Thus, cutting the threads is essential though it leads to significantly prolonged surgery duration. Self-cutting screw systems (e.g., TACKERt; Inion Ltd. Tampere, Finland) can be considered as an efficient alternative, though featuring a high fracturing risk due to torsional forces applied on the junction between the screw head and neck [70].

For these reasons, for example, Eckelt and coworkers [71] used the SonicWelds system developed by KLS Martin (Tuttlingen, Germany) in eight patients with craniosynostosis, after successful application in animal experiments [72]. In this system, the osteosynthesis Resorb-X® material is fixed by inserting resorbable pins through ultrasounds (bone welding), reducing the time required for applying the osteosynthesis materials by about 50% [72]. Thus, ultrasound aided fixation using resorbable osteosynthesis materials resulted more stable than screw fixation, due to fixation in both cortical and cancellous bone. The patients were followed up for 12 months and pin fixation was stable in all cases [72].

On the basis of the clinical experiences with the biodegradable plates and screws, it is evident that they can be considered a promising alternative to metallic plating systems in paediatric CS patients, due to their very low complication rates (wound infection and palpability of implant of less than 1%). However, their successful implantation strongly depends on the operator's experience and his/her level of expertise in cranioplastic remodelling with rigid fixation systems [54].

**Table 3.** Commercial rigid fixation systems based on biodegradable materials.

| Commercial Fixation System | Composition | Post-Operative Infection Rate | References |
|---|---|---|---|
| LactoSorb® | 82% poly-l-lactic acid (PLLA), 18% polyglycolic acid (PGA) | 0–3% | [54–56,58–62,73] |
| Biosorb® PDX | 80% PLLA 20% PGA | 4% | [63] |
| PolyMax® RAPID | 85% PLLA, 15% PGA | None | [74] |
| Inion® CPS baby | Trimethylene carbonate, PLA, PDLLA, Polyglycolide | 0–1.6% | [65–68] |
| RapidSorb® | 85% PLLA, 15% PGA | 0–1.4% | [75,76] |
| MacroPore® FRP | 85% PLLA, 15% PGA | 2–4% | [55,64] |
| Resorb-X® | 50% PLLA, 50% PDLLA | 0–2.6% | [77,78] |

*2.2. Tissue Engineering Strategies*

Despite significant innovation in the management of CS, morbidity and mortality still exist. Residual cranial defects represent one of the potential complications. In fact, all the surgical correction techniques rely on spontaneous ossification of the surgical bone defects. In spite of a proper preservation of *dura mater* and periosteum, critical size cranial defects may result in a large proportion of cases, from incomplete or defective spontaneous healing of the bone [79] or other surgical complications (i.e., infection or resorption of the bone flap) especially when multiple repeated surgeries are required, as in the case of re-synostosis [80].

The resulting cranial defects may represent a serious issue in paediatric neurosurgery [81,82]. The common procedures adopted in these cases involve the use of either autologous bone grafts, or allogeneic grafts, or alloplastic materials to fill the bone gap. The use of autologous bone and avoidance of heterologous material is pursued, since an ideal bone substitute is lacking and complications of heterologous materials are higher in children [46]. Cranial bone splitting may be considered for small to medium-size defects [83]. In large size (>5 cm$^2$) defects, the options are very limited.

Biobanked bone-derived products, such as demineralized bone matrix (DBM) and bone dust (BD), represent potential alternatives to autologous bone. A recent study assessed the outcomes of DBM plate implantation, with or without bone dust, for the treatment of large calvarial defects resulting from cranial vault surgery in CS patients above one-year-old [74]. Their results indicated a statistically significant improvement of bone healing of calvarial defects receiving DBM plus BD compared with patient-matched control defects, without any complication, particularly in older patients [84].

Certainly, in the last years, the introduction of custom-made implants has warranted better esthetical outcome and reduced complications with respect to synthetic materials that were molded intraoperatively, namely, bone cements and polymethyl methacrylate (PMMA). However, the long-term outcome of the heterologous materials is poorly documented through the literature, though most of them have shown good results in the short-term period. This is the main reason that has prompted the research of biomimetic materials, to improve osteointegration and reduce complications.

The most exploited materials for traditional adult skull reconstruction (i.e., PMMA, PMMA hybrid cements, titanium, polyetheretherketone (PEEEK), and mesoporous hydroxyapatite (HAp)), present significant limitations that make them poorly suitable for treating the growing skull of paediatric patients [85–89].

Porous HAp is widely used as material for cranial defects in paediatric patients, due to its similarity to the inorganic component of the bone extracellular matrix [90,91], yielding limited foreign body reactions and excellent cosmetic results. Nonetheless, it presents several drawbacks, including the brittle nature, the low tensile strength, and the high infection rates [92]. Cranial implants based on macroporous hydroxyapatite have been increasingly used with satisfactory results in children [93,94]. However, these implants are not indicated under two years of age, so that a real solution for the repair of large

size cranial defects in this age group is actually still lacking. The cranial repair in these cases require a careful approach considering the need to accommodate the still growing brain in order to avoid secondary constriction and a failure of the cranioplasty. Exchange cranioplasty may represent an effective option. This consists of harvesting a graft from the unaffected skull to repair the contralateral cranial defect, and relying on the spontaneous healing of the donor site if the dura mater and periosteum are well preserved [81,95].

Tissue engineering has been introduced in CS care, to cope with the severe drawbacks associated with the use of "non-growing", non-biological metallic and bio-ceramic-based implants, yielding poor and unsatisfactory long-term results in paediatric patients [46].

The tissue engineering approaches involve the combination of three main components, i.e., scaffold, biomolecules (e.g., growth factors and bone-inducing agents, drugs, such as antimicrobials, etc.), and cells, in order to fulfil the key requirements of an ideal bone regenerative strategy: osteoconduction, osteoinduction, and osteogenesis [96].

The design and development of temporary biomimetic scaffolds is pivotal in determining the success of a tissue engineering approach. The ideal scaffold should be biodegradable, able to promote the bone regeneration by supporting cell colonization and growth (osteoconduction), and/or to induce stem cells to differentiate towards the osteoblast phenotype (osteoinduction). Hence, two different strategies can be followed: (i) a conductive approach based on the use of passive three-dimensional supports on which cells may attach, migrate, and differentiate; (ii) an inductive approach, based on the use of active supports, loaded with bioactive signals, aimed at promoting cell migration and guiding the cell responsiveness.

To these aims, the designed scaffolds have to satisfy multiple requirements, in terms of chemical-physical properties (composition, microstructure, mechanical properties, bioresorption degree) and biological features (biocompatibility, ability to promote cellular and vascular colonization, angiogenesis, etc.) [97].

Biodegradable materials, both natural and synthetic, provide many advantages over metals and other non-degradable materials commonly used in maxillofacial surgery, avoiding the related disadvantages, i.e., growth disturbance [98], plate migration [99], the need for subsequent removal [100], long-term palpability and thermal sensitivity [101], and compatibility with imaging investigations.

However, natural polymers, e.g., collagen, alginate, agarose, chitosan, and fibrin, present poor mechanical behaviour and inconsistent degradation rates, whereas the synthetic polymers, i.e., polycarbonates, polyesters, polyorthoesters, polyanhydrides, polyurethanes, and polyphophosphazenes [102], are characterized by higher mechanical strength.

In detail, the physical (i.e., mechanical and degradation) properties of polymers strongly depend on their molecular weight, crystallinity degree, physical aging time, test, and environmental conditions. For example, PLA presents relatively high mechanical properties (flexural strength up to 140 MPa, Young's modulus 5–10 GPa, total tensile elongation about 3%, Charpy impact fracture ~2.5 kJ/m$^2$), with excellent optical properties, good processing ability (with low shrinkage not causing product deformation) and biodegradation time over the period of several months up to two years [103]. On the other hand, PCL is a biodegradable polymer with very high flexibility, a tensile strength between 4 and 785 MPa and a Young's modulus in the range 0.21–0.44 GPa, presenting a degradation time of 2-to-3 years [104].

In particular, to contain/avoid the recurrence of suture fusion following surgical resection (re-synostosis) in CS patients, innovative approaches could derive from the design of biomaterials able to tame tissue mineralization.

The combination of biomaterials with stem cells and osteoprogenitor cells allows implementing the osteogenicity in the tissue engineering approach. Nonetheless, cell-based treatments involve additional drawbacks, mainly ascribable to immune-related issues, along with increased production costs. On the other hand, the use of autologous cells to overcome the risk of immune rejection implies additional morbidity at the tissue

harvesting site (bone marrow, adipose tissue, etc.), and hampers their exploitation in younger patients, for whom a real clinical translation of advanced cell-based therapies for CS treatment cannot be foreseen.

Different studies tested the feasibility of tissue engineering approaches for the treatment of membranous bone critical-size defects, relying on either cell culture systems or animal models (see Table 4). In particular, Cowan and coworkers [105] produced apatite coated poly-lactic-co-glycolic acid (PLGA) (PLA:PGA 85:15) scaffolds through solvent casting and a particulate leaching process, and seeded them with adipose-derived stromal cells (ASCs) or bone marrow stromal cells (BM-MSCs). They evidenced a remarkable intramembranous bone formation by 2 weeks and areas of complete bony bridging by 12 weeks, after in vivo implantation in adult male FVB mice (FVB is an albino, inbred laboratory mouse strain that is named after its susceptibility to Friend Leukemia Virus B), suggesting that ASC cells are able to repair critical-size skeletal defects without genetic manipulation or the addition of exogenous growth factors [105]. This supports the idea that the microenvironment at the bone defect site could induce the osteogenic commitment of extraskeletal MSCs, such as ASCs, on appropriate scaffolds promoting efficient bone healing in vivo [106,107].

A recent study lead by Yu et al., demonstrated the efficacy to restore coronal suture patency in a Twist1$^{+/-}$ mice Model using Gli1$^+$ MSCs combined with methacrylated gelatin (GelMA) modified with Matrigel and collagen I (COL-I) [108]. GelMA (M-GM) scaffold is highly biocompatible and biodegradable, and it easily adapts to defects [109]. The Authors showed that loss of Gli1$^+$ MSCs induces premature coronal suture fusion in Twist1$^{+/-}$ mice, confirming that this subpopulation of MSC are largely required for craniofacial bone turnover, homeostasis and repair [110]. They also showed that the implantation of MSC-graft leads to suture regeneration by restoring Gli1$^+$ cells subpopulation within cranial suture both through exogenously implanted Gli1+ MSCs and endogenous MSCs derived from the *dura mater* [108]. Our group has indeed recently demonstrated that GLI1 represents a specific marker for MSC in the human calvarial niche [21].

Additional biomaterials have been tested as suitable scaffolds for CS skull reconstruction, by studying their interaction with cells in vitro. In particular, a scaffold composed of a bioactive glass and a bioabsorbable 80:20 L-lactide:glycolide copolymer (PLGA 80) was developed and tested with both murine osteoblast cell lines expressing either the wild type or the mutated FGFR2 (namely, the FGFR2-C278F mutation found in Crouzon syndrome), and with human primary osteoprogenitors from syndromic and nonsyndromic CS patients [111]. The composite scaffolds were able to support the homing, adhesion, and differentiation of both normal and mutated osteoprogenitor cells, hence, proposed by the Authors as a suitable bone substitute to be implemented in the care of CS patients, undergoing extensive reconstructive surgery [111].

A revolutionizing approach has been introduced with the additive manufacturing techniques, which enable the precise and customized replication of the architecture of bone defects from medical image data (computerized tomography). In this way, the morphology of the synthetized scaffold perfectly matches the defect to be filled, hence facilitating its surgical implant and graft retention [112,113].

A 3D scaffold based on polyethylene glycol (PEG) hydrogel-coated polycaprolactone (PCL) was developed using a novel computer-aided precision extrusion 3D printing system [114]. PCL shows excellent solubility, low melting point, biocompatibility and easy manufacturing [115–118], while the PEG coating enables inhibiting osteoblast differentiation [119]. Indeed, in the cited study MC3T3E1(C4) calvarial cells adhered and differentiated into osteoblasts only on the uncoated portion of the scaffold [114].

Another group developed a new 3D-printed β-tricalcium phosphate (β-TCP) scaffold loaded with the osteogenic agent dipyridamole and evaluated the effects of this implants to support bone growing within bilateral calvarial defects in rabbit [120]. After the implantation, the Authors observed a volumetrically significant osteogenic regeneration of calvarial defects, with a favorable preservation of suture patency, at least in the short term [120]. The

same group then reproduced the experiments in an immature rabbit model, evidencing a comparable responsiveness with respect to autologous bone grafts. They revealed a volumetrically and functionally significant osteogenic regeneration of calvarial defects, with a neoformed vascularized bone comparable to native tissue. Moreover, the Authors confirmed, using 3D morphometric facial surface analysis, that the 3D-printed β-TCP and dipyridamole scaffold does not lead to premature closure of sutures and allows to maintain the normal craniofacial growth [121].

Similar conclusions were reported by Bekisz and coworkers [122], who tested the same scaffold in sheep calvarial defects, and observed no exuberant or ectopic bone formation, and no histologic evidence of inflammation within the defects, and higher osteogenesis in vivo.

In the case of cranial defects characterized by irregular shape the use of injectable scaffolds is desirable, allowing to completely fill in the created void and to avoid invasive surgery [123], hence reducing the associated morbidities and costs [124]. The commonly used injectable scaffolds are based on hydrogels that consist of three-dimensional polymeric networks able to absorb a large amount of water, while maintaining their structural integrity. Hydrogels are widely used for many biomedical applications, such as scaffolds for tissue regeneration [125], cell encapsulation [126], drug delivery [127], and bioadhesives and biosealants [128]. Indeed, hydrogels can be designed to be responsive to environmental changes (e.g., temperature, pH, and ion concentration) and to encapsulate functional biomolecules and nanoparticles [129]. Shear thinning injectable hydrogels present several advantages over other systems, owing to their higher defect margin adaptability, easier handling and ability to be manually injected into deeper tissues. Shear thinning hydrogels consist in ex vivo pre-formed hydrogels that are delivered in vivo applying shear stress during injection (commonly through a syringe) and quickly self-heal after shear removal [130]. Conversely, in situ hydrogels either require a cross-linking agent (often toxic), or exploit physical properties (e.g., temperature, and pH), to transit from sol to gel upon being injected into the defect site. Therefore, the in vivo environment can affect the crosslinking behavior of in situ gelling agents, while it has a negligible effect in the case of shear thinning hydrogels [131].

PLGA-based colloidal gels, produced using poly(ethylene-co-maleic acid) (PEMA) and polyvinylamine (PVAm) as surfactants, loaded with dexamethasone (DEX), have been exploited in rat cranial bone defects treatment. The tested hydrogels supported osteoconduction, promoting bone formation, whereas the untreated cranial defects showed negligible bone formation and collapsed [132]. Furthermore, a very low and delayed DEX release was achieved over two months from the loaded the PLGA nanoparticles and for one month when DEX was blended with the particles [132]. Another group developed a calcium sulfate ($CaSO_4$) and FGF-18 (Fibroblast Growth Factor-18) loaded chitin–poly(lactide-co-glycolide) (PLGA) composite hydrogel for rat craniofacial bone defect regeneration. They revealed a sevenfold increase in the elastic modulus compared with the neat chitin–PLGA hydrogel, a sustained release of FGF-18, an enhanced alkaline phosphatase (ALP), increased endothelial cell migration, early and almost complete bone healing in comparison with chitin–PLGA/$CaSO_4$, chitin–PLGA/FGF-18, chitin–PLGA, and sham control system, respectively [133].

### 2.3. Bioactive Compounds and Delivery Systems

Several studies of deregulated pathways caused by mutations underlying craniosynostosis have provided promising compounds in designing non-invasive adjuvant treatments for CS patients. On this regard, Bai and colleagues showed that recombinant mouse periostin can reduce proliferation, migration and osteogenic differentiation of suture-derived cells, as well as can decrease coronal suture fusion in Twist1$^{+/-}$ mice model of Saethre–Chotzen syndrome, restoring the loss of TWIST1 due to haploinsufficiency [134].

**Table 4.** Tissue engineering approaches for bone regeneration in calvarial defects.

| Type of Polymer(s) | Synthesis Procedure/ Scaffold Assembly | Molecular Functionalization | Cell Types Implemented | Testing Model: *In Vitro* | Testing Model: *In Vivo* | Reference |
|---|---|---|---|---|---|---|
| PLGA (PLA:PGA 85:15) | Solvent casting + particulate leaching | - | ASC/ BM-MSC/ Calvarial osteoblasts/ dural cells | - | FBV mice | [105] |
| GelMA + Matrigel + COL-I (M-GM) | Gelatin with methacrylamide (GelMA) side groups cross-linked by radical polymerization via photoinitiation mixed with Matrigel and COL-I | - | Gli1 + MSCs | - | *Twist1*+/ mice | [108] |
| Bioactive glass + PLGA (PLA:PGA 80:20) | Fiber assembly scaffolds | - | - | Transgenic FGFR2$^{C278F/wt}$ murine osteoblasts; osteoprogenitor cells from CS patients | - | [111] |
| PEG hydrogel-coated PCL | CAD extrusion 3D printing | - | - | MC3T3E1(C4) murine calvarial MSCs | - | [114] |
| β-TCP scaffold | Custom-built, direct-write 3D printing | dipyridamole | - | - | Rabbit/ sheep calvarial defects | [120–122] |
| PLGA nanoparticle | Solvent diffusion method | DEX | - | - | Sprague-Dawley rats | [132] |
| Chitin-PLGA composite hydrogel | Chitin regeneration technique | CaSO$_4$ and/or FGF-18 | - | Rat adipose derived stem cell (rADSCs) | Sprague-Dawley rats | [133] |

Other approaches have evaluated the feasibility of pharmacological therapies acting through interference or downregulation of FGF/FGFR2 and Wnt signalling at the suture interface. Indeed, Shukla et al., demonstrated the possibility of preventing Apert-like phenotype in mice by targeting the dominant mutant form of *Fgfr2* with a small hairpin RNA [135]. This was recently confirmed in another study, using small interfering RNAs (siRNAs) targeting the Apert mutated Fgfr2 allele were used to inhibit osteoblastic differentiation and matrix mineralization, by reducing the signaling of ERK1/2 and P38 in vitro (cultured patients' primary calvarial cells) and ex vivo (calvarial explants from Apert mice) [136].

Furthermore, the development of mouse model of CS demonstrates that early suture ossification could be rescued through selective attenuation of docking protein Frs2-α, which recruits a variety of adaptor proteins upstream the RAS/MAPK/ERK pathway [137].

Tested strategies include also inhibitors of FGFR2 tyrosine kinase (PD173074), Wnt/β-catenin, MEK1 and 2/ERK [135,138] pathways.

Additional potential molecular targets for the development of innovative treatments for CS have been tested within the FGF signaling. In a fibroblast growth factor-18 (*Fgf18*)-deficient mice, generated through gene editing, proliferation and osteogenesis of calvarial mesenchymal cells were decreased, and suture closure was delayed [139]. Furthermore, Quarto and collaborators observed that fibroblast growth factor 2 (FGF-2) inhibited the

osteogenic differentiation of adipose tissue-derived stromal (ADS) cells in a dose-dependent and reversible manner [140]. Pitfalls may pose criticisms, as the inability of bio-compounds to be appropriately and stably delivered by scaffolds, due to short-half life, poor tissue penetration, instability/lability, and large molecular size. To overcome such limitations, the research has pushed the development of molecular "carriers" able to deliver and to maintain the bioactive compounds in the suture site. The efficacy of vehicles is dependent on the type of material, the biocompatibility, the non-toxicity, the biodegradability, the encapsulation, and the concentration of biomolecules incorporated within the material, along with the release kinetics [141] (see Table 5).

Rapid developments in nanotechnology and controlled drug delivery have triggered exceptional growth in treating various bone diseases [142]. Most bone tissue engineering approaches rely on the implementation/functionalization of osteoconductive scaffolds with bioactive compounds, able to modify host tissue homeostasis upon grafting.

Several studies have reported that the size variation of drug carriers in the nanoscale range (1–100 nm) provides various advantages for drug delivery purposes, such as enhanced transport across cell membranes, thus reducing clearance from the body and providing a selective targeted drug delivery; greater surface area-volume and subsequently more surface reactivity, thus increasing drug loading ability, providing controlled dissolution rates and drug bioavailability; and size similarity to natural tissue components, thus enabling better tissue acceptance by biomimicking tissue architecture [143–146]. Moreover, the inherent properties of nanoscale materials (such as physical, chemical, mechanical, electrical, magnetic, and optical properties) can be utilized to improve the performance of the delivery systems [147–149]. For example, electrical properties of surfaces interact with drugs or biological systems to promote an even greater impact on drug biological activity, drug release kinetics, conjugation to targeting moieties, and transport in bone. For instance, cationic nanoparticles can localize in the cytoplasm and within mitochondria, while anionic nanoparticles remain in lysosomes [150].

These novel drug delivery systems involve different types of materials from 1D to 3D: polymers, metals, ceramics, semiconductors, and sol–gel with different geometries including particles, fibres, capsules, tubes, whiskers, and dendrimers [146].

Some of these delivery systems have been exploited for the confined administration of therapeutics in in vitro and in vivo models of CS.

Collagen has been used in the form of films, hydrogels, pellets, shields, and sponges, for the delivery of specific inhibitors of the aberrant pathways involved in the molecular pathogenesis of CS [151,152]. The collagen gel is an attractive vehicle because it can be easily inserted into a variety of spaces, it is extremely malleable, and can be introduced in a minimally invasive manner, though the release kinetics is variable, and often transient (i.e., lasting only a few weeks) [141].

The Bone Morphogenic Protein (BMP) signalling pathway represents a suitable target for the selected delivery of specific drugs able to interfere with the osteogenic program, being involved in the differentiation, both in syndromic and nonsyndromic CS [38,153]. BMPs are members of the transforming growth factor beta (TGFβ) super-family and are potent osteo-inductors [154]. Different studies have tested the role of BMP antagonists, namely, noggin, glypicans, or gremlin, for developing adjuvant treatments in CS management [138,153,155,156]. Specifically, noggin loaded in a collagen-vehicle [155] or on a gel-foam scaffold with GFP-expressing cells [138] was administered topically at the site of suturectomy. This clearly reduced the re-ossification rate within pathological sutures in treated animals compared with untreated controls, demonstrating that the inhibition of post-operative re-synostosis was possible using biologically based therapies [138,155].

More recently, Premaraj and colleagues used a plasmid encoding TGFβ3 delivered by a dense collagen-gel scaffold injected at suture sites in rats, to prevent programmed suture fusion in calvarial organotypic culture. The treatment enabled a 70% to 80% folds decrease of suture ossification compared with collagen controls, depending on suture sites [157].

Collagen sponges were used in an in vivo experiment of Hong and Mao [158]. The researchers supposed that skull suture can be engineered from autologous cells. In particular, the Authors suggested that re-synostosis in CS patients occurred lacking mesenchymal and fibrous interface between the gap created after craniectomy. They fabricated a composite tissue construct made by fibroblasts isolated from rabbits seeded in an absorbable gelatin-scaffold between two collagen sponges loaded with recombinant human BMP2 (rhBMP2). Surgically created full-thickness parietal defects were filled with the composite tissue implant in the same rabbits from which fibroblasts had been obtained. After four weeks of in vivo implantation, the treated cranial suture was still patent [158]. However, the biological introduction of rhBMP2 needs further studies due to the risks of developing premature suture fusion, osteolysis and malignant degeneration [159].

Nonetheless, some studies have shown that a few types of collagen-based vehicles are able to promote osteogenesis in rat cranial defects, making them unsuitable for bone inhibiting applications [141,160,161].

PLGA polymer is a primary candidate in regenerative medicine due to its biocompatibility, controlled structural and mechanical proprieties and tailored degradation rates, being also suitable for growth factor delivery. PLGA may be fabricated in various forms, including sheets, blocks, microspheres, and nanofibers [162–167]. Given the long-term need for cranial suture regulation throughout the development of the skull in early childhood, PLGA microspheres offer great advantage in comparison with other release systems, due to their potential for prolonged release as well as injectable dimensions [168].

A demonstration of biological replacement of a complex anatomic structure in an in vivo model using autologous cells and drug delivery approach consisting of PLGA microspheres was reported by Moioli and coworkers [166]. The researchers used autologous BM-MSCs co-seeded in a collagen carrier with TGFβ3 encapsulated within PLGA microspheres in order to generate a cranial suture engineered tissue. The construct was applied during craniectomy in the same rat that had earlier donated the bone marrow sample. The analysis demonstrated a biologically derived bone-soft tissue-bone interface compared with the ossified suture derived after the treatment without autologous BM-MSCs. This type of scaffold in CS would be advantageous for both promoting formation of skull suture tissue and inhibiting the fusion of the suture [169,170].

Other polymers may be used to form blend materials of optimized properties, such as improved mechanical strength, and defined degradation rates, i.e., PEG-PLGA composites, PLGA-Poly(isoprene) PI, or PLGA-PCL [171]. These polymers may be also used in the absence of PLGA to produce alternative systems able to release desired bioactive compounds. An interesting study by Hyzy and coworkers [172] employed a PEG based hydrogel containing anti-angiogenic compounds such as anti-vascular endothelial growth factor A (VEGFA)-antibody or hypoxia inducible factor 1α-inhibitor topotecan. The Authors were able to demonstrate that these compounds can be incorporated into a spontaneously polymerizing hydrogel and remain active over 14 days in both in vitro and in vivo murine models. Specifically, bone formation was delayed by inhibiting neovascularization, suggesting a possible use as a therapeutic approach to control re-synostosis following suturectomies where rapid osteogenesis is not desired [172].

A further study investigated an injectable in situ crosslinking hydrogel composed of two mutually reactive poly(ethylene glycol) macromolecules for controlled delivery of Gremlin1, an inhibitor of BMPs, in order to treat cranial defect of weanling mice. The Authors demonstrated that the hydrogel with Gremlin1 was able to delay, for up to 14 days, the rapid post-operative bone growth that occurs within the pathological suture. However, the results from the long-term study showed that this therapeutic strategy was unable to completely prevent the re-synostosis over a long period and therefore would require repeated injections or changes in the kinetic release of bioactive molecules [173].

Recently, Bariana and coworkers studied the effect of glypicans 3 (GPC3) released by titanium nanotubular implant (TNT/Ti) to prevent re-synostosis in a murine model of Crouzon syndrome [174]. GPC3 is an antagonist of BMP pathways with a long-term

potential in controlling post-operative suture ossification in CS patients, compared with noggin [141,156,175,176]. The system delivery was fabricated with self-ordering electrochemical process and has some specific advantages, namely, outstanding structural properties (mechanical strength), excellent biocompatibility, high in vivo stability, nontoxicity, and tailorable drug loading. These properties enable controllable therapy that does not require surgical removal after implantation [177]. Furthermore, chitosan coating may be used to extend the duration of drug elution and to improve biocompatibility in orthopedic implants [129,147]. Accordingly, TNT/Ti nano-implants coated with chitosan were applied in a cranial defect created in Crouzon mice. The implants stably adhered and were preserved up to 90 days after surgery, without any adverse reaction. The pathological suture was still unfused in the site of the implants highlighting the success of the nanotubes to preventing re-ossification [178].

Yokota and co-workers developed a purified soluble form of FGFR2 harboring the S252W Apert syndrome mutation truncated at the extracellular domain (sFGFR2IIIc$^{S252W}$), delivered by a nanogel [179]. This modified protein was able to bind Fgf2 and to dimerize with the full-length forms of FGFR2 (FGFR2IIIc$^{S252W}$ or FGFR2IIIc), resulting in incomplete dimers that inhibited downstream signalling in embryonic calvarial sutures derived from Apert syndrome mice [179]. The experiments showed that treated coronal sutures remained patent compared to the untreated ones. The cholesteryl-bearing pullulan (CHP) nanogel used in this study was composed by hydrophilic polysaccharides partially modified with hydrophobic cholesteryl groups with the addition of acryloyl to PEG containing four branched terminal thiol groups [179]. This compound self assembles in water and forms stable nanogels with a diameter of 30 nm and has two unique characteristics: a high loading capacity for biomolecules inside their nanospaces and a chaperone-like activity able to deliver specific bio-compounds in the target site [180–182]. The advantages of a nano-delivery based system as protein carrier are multiple, such as storing proteins and gradually local releasing, biocompatibility, and cheapness. Nonetheless, biomaterials that have been explored so far showed undesired pharmacokinetics and uncontrolled release patterns, and/or are too complex [141,170].

Finally, phytochemicals, phenolic components extracted by plants, have been also used for bone regeneration, owing to their high availability, low cost, reduced toxicity, and excellent inherent biological properties that make them promising alternatives to synthetic growth factors and cytokines [183,184]. On this regard, a phytochemical-reinforced laponite hydrogel bone sealant was tested in the repair of non-healing murine cranial defects [185]. The tested compound was obtained via the self-assembly of phytochemical-grafted chitosan (PGC) with laponite, involving the phytochemicals catechol groups, which yielded a malleable hydrogel with improved mechanical, antibacterial, antioxidant, and osteo-inductive properties, to be injected into defects with complex geometries [185]. The antimicrobial and antioxidant properties resulted from the phenolic hydroxyl-rich moieties of the phytochemicals and laponite allowed to favor the osteo-inductivity by regulating the Wnt/$\beta$-catenin pathway, and acting as a nanocarrier for controlled drug delivery. Another study investigated the therapeutic effects of caffeic acid phenethyl ester (CAPE), a component of honey bee-hive propolis with antioxidant, anti-inflammatory, antiviral, and anticancer properties, associated to β-tricalcium phosphate/hydroxyl apatite particles, in a rat critical size cranial model, obtaining improved bone defect healing [186].

**Table 5.** Bioactive compounds and delivery systems.

| Delivery System Chemistry | Formulation | Bioactive Molecule(s) | Target Cells/ Compartments | *In Vitro* Testing | *In Vivo* Testing | Reference |
|---|---|---|---|---|---|---|
| Collagen | Slow-resorbing collagen vehicle | Noggin | Calvarial bone cells (suturectomy site) | - | white rabbits with bilateral coronal suture synostosis | [155] |
| Collagen | Gelfoam scaffold | Noggin | Calvarial bone cells (suturectomy site) | - | C57BL/6J mice | [138] |
| Collagen | Gel-like scaffold | TFβ-3 expression plasmid | Cranial suture | Rat calvarial organ culture | Sprague-Dawley rats | [157] |
| Collagen | Gelatin-microporous sponge scaffold | rhBMP2 and dermal fibroblasts | Cranial suture (parietal defect) | - | New Zeland white rabbits | [158] |
| Collagen— PLGA 50:50 copolymer | Microspheres | Murine BM-MSCs or human BM-MSCs + TFβ-3 | Cranial suture | culture of human BM-MSCs and BM-MSCs-derived osteoblasts | Rat craniosynostosis model | [168,169] |
| DB-co-PEG/poly(TEGDMA)-co-(N3-TEGDMA) | Hydrogel | anti-VEGFAantibody and topotecan | Endothelial cells (posterior frontal suture) | Human aortic endothelial cell cultures | C57Bl/6J mice | [172] |
| PEG | Hydrogel | rmGremlin1 | Cranial suture (posterior frontal suture) | MG63 cells | Murine model C57Bl/6J | [173] |
| TNT/Ti | Nanotube | GPC1 or 3 | Cranial suture | C2C12 murine myoblast cell line | Crouzon murine model (Fgfr2$^{c342y/+}$) | [177,178] |
| CHP composed by hydrophilic polysaccharides partially modified with hydrophobic cholesteryl groups additioned with acryloyl | Nanogel | sFGFR2IIIc$^{S252W}$ | Cranial suture (coronal suture) | Calvarial coronal suture cells of Apert Syndrome mice | - | [179] |
| PGC + laponite | Nanocomposite hydrogel (NC–organic hydrogel bone sealant NoBS) | Laponite+ Smoothened agonist (SAG) | Calvarial bone defect | Mouse BMSC line (D1 cell, CRL-12424) | CD-1 mice | [185] |
| β-tricalcium phosphate/ hydroxyl apatite | Particles | CAPE | Calvarial bone defect | - | Sprague-Dawley rats | [186] |

## 3. Cell-Based Disease Modelling: From 2D to 3D Culture Systems

Advances in understanding the biomolecular mechanisms involved in suture fusion may allow the development of adjuvant therapies aimed to minimize complications linked to cranial vault remodelling, like re-ossification of the pathological suture. To date, different disease models have been exploited to study CS etiopathogenesis and pathophysiology,

including transgenic mouse models for syndromic CS (reviewed by [187]). Nonetheless, considering the wide heterogeneity of CS etiology (see introductory paragraphs and Tables 1 and 2), appropriate generalized disease models cannot be developed, especially for nonsyndromic cases. In addition, significant differences exist between humans and mice in the overall architecture and development of the skull, including early postnatal murine lethality of some CS mutations [155]. Taken together, these observations explain the need to exploit somatic cells derived from CS patients' calvarial tissues obtained from surgical wastes. In particular, our group studied the molecular mechanisms implicated in the overactive osteogenic cascade of mesenchymal stromal cells isolated from calvarial tissues of nonsyndromic CS patients, with idiopathic etiology. By studying this cellular model, we identified functional biomarkers (LMP3, BBS9, RUNX2, AXIN2, and GLI1), to be exploited also in the design of molecular targeted therapeutic approaches aimed at regulating the osteogenic commitment of stem cells in the suture niche [10,12,15,21]. Furthermore, Barreto and coworkers developed a 2D culture system based on cells isolated from nonsyndromic CS patients' sutures, to demonstrate that fused-vs-patent suture cells display differential gene expression profiles underlying different stiffness-mediated responses [188]. This evidence proves that the microenvironment influences the mechanotransduction signalling ultimately affecting the osteogenic phenotype, and leads to premature suture fusion [188]. Interestingly, this molecular signalling includes the BBS9-related primary cilium activation cascade found dysregulated in calvarial cells isolated from naturally occurring nonsyndromic CS [10].

However, some critical limitations should be considered when using bidimensional homogeneous cell culture models, including the difficulty to study the cell–cell and cell–environment interactions, as they cannot mimic and recapitulate the heterogeneity and complexity existing within tissues in vivo. These concerns are being overcome by the advent of induced pluripotent stem cells (iPSC), combined with the rapid development of 3D culture models.

On one hand, iPSC technology allows reprogramming patient-specific somatic cells (e.g., skin fibroblasts and blood cells of patients with a monogenic disorder, such as syndromic CS) towards a pluripotent state by defined exogenously administered factors [189,190]. Once obtained as stable cultures, iPSCs can be induced ex vivo to effectively differentiate towards the diversified cell lineages existing within living tissues, establishing a personalized disease model that faithfully recapitulate the hallmark of disease pathophysiology [191]. iPSCs are widely exploited in disease modelling and drug testing, by establishing, in 2D cultures, 3D models, organoids, and human–rodent chimeras [191]. The combination of iPSC with genome editing techniques further boosted the field of personalized disease modelling, enabling the introduction of gene mutations into wild-type cells to study the molecular pathogenesis of disease, and their effects on cellular differentiation, tissue development and morphogenesis [192]. In the study by Matheus and colleagues, iPSC of patients suffering from Bohring–Opitz syndrome, a complex developmental disorder entailing CS, caused by mutations in the ASXL1 gene, were produced to obtain neural crest (NC) progenitors xenotransplanted into chicken embryos. The Authors used this chimeric model to study the molecular pathophysiology of the disease and demonstrated an impairment of NC delamination and emigration during early neurocranial development [193].

On the other hand, 3D culture systems, based on either somatic cells or iPSC-derived cells cultured on appropriate biomaterials serving as scaffolds, are further improving the development of patient-customized models of mendelian and/or complex disorders, such as CS. On this regard, Yang and colleagues developed a 3D culture system in which osteoblasts collected from the long bone of mouse model of Apert syndrome were homogeneously encapsulated in a poly(ethylene glycol)-diacrylate (PEGDA) 3D hydrogel [194]. Various works in literature demonstrated the importance of physical three-dimensionality of the matrix in regulating osteoblast behaviour, including cell osteogenesis and bone matrix formation (see [195] for an up-to-date review). Indeed, the Authors demonstrated that

the expression of collagen type-I and -II and osteocalcin was higher, whereas the levels of matrix metalloproteases and the BMP inhibitor Noggin were lower in mutated osteoblasts encapsulated in 3D scaffolds compared with control cells [194]. This allowed confirming that the Apert Fgfr2 mutation was associated with increased osteogenesis, abnormal chondrogenesis, as observed in vivo. Therefore, the direct correlation between in vitro and in vivo findings supports the use of 3D culture systems as valuable alternative to animal models, though limited to very preliminary stages of preclinical disease modelling.

## 4. Clinical Perspectives and Actual Clinical Translation

Even if important advancements have been achieved in adult craniofacial bone tissue engineering [196], an optimization for use in the pediatric population has not been reached yet [197,198]. A personalized approach is needed for treating pediatric patients to take into account multiple features: the reduced bone thickness [199], the decrease of *dura mater*'s osteoinductive potential after 12 months [200], and the dynamic growth and development of the craniofacial skeleton requiring multiple surgical treatments [201]. Thus, an ideal pediatric bone replacement would satisfy a lot of requirements to re-establish the functionalities of the skeleton without avoiding craniofacial development [198,199]. This is particularly relevant in the treatment of congenital bone developmental defects. In craniosynostosis care, cranial surgery for bone reconstruction should concurrently cope with the need to hamper premature suture fusion, to allow the correct pace for braincase enlargement along with the harmonic craniofacial development.

In the present review, the advantages of the surgical approaches were compared, in terms of both topology of the suture involved and of the possible clinical strategies to be implemented.

Although several tissue engineering approaches have been designed and tested in preclinical studies, it is evident that, regardless of the surgical approach, an optimized device/scaffold to promote the calvarial bone regeneration, while simultaneously preventing the excessive bone formation at the site of suturectomy, thereby avoiding re-synostosis, has not reached an universal consensus to be stably translated in the clinical practice. The use of additive manufacturing techniques for a customized replication of the architecture of the skull and bony defects, based on tomographic data, would facilitate the achievement of this ambitious purpose, promoting surgical placement and retention of the engineered scaffold. Therefore, based on such considerations, the application of bone tissue engineering strategies in this specific pediatric context is limited and at an infancy state [202]. Nevertheless, the everlasting efforts of the scientific community have achieved some specific advancements in the field of novel biomaterials and drug delivery systems. A unique gold standard polymer or compound cannot be universally defined, also due to the wide and heterogeneous spectrum of CS entities (Tables 1 and 2), offering a correspondingly wide range of outcomes and suggesting the need for personalized strategies. On the other hand, a consensus seems to emerge from the extant literature reviewed in this paper: injectable gel formulations are probably the more suitable to be implemented as a tissue engineering approach in CS surgery (Table 5). To achieve a feasible personalized bone regenerative treatment for CS, novel data are expected to arise from the development of 3D human culture systems, able to mimic the patient-specific in vivo environment, as platform for improved ex vivo preclinical testing.

Finally, given the crucial aspect of timing in CS surgery, the current pandemic situation that has led to delay of elective surgeries, posed CS care in a critical position, at least in selected healthcare systems, worldwide. Indeed, postponing the schedule of CS surgical treatment may cause increased risk of disease progression, including enhanced intracranial pressure and completion of suture fusion, hence raising the demand for more invasive and higher risk procedures, such as open vault surgery [203]. This aspect could reasonably emerge as an additional Covid-19 side effect affecting pediatric patients, and will further prompt the improved implementation of bone reconstructive strategies and personalized tissue engineering approaches in craniofacial surgery.

**Author Contributions:** Conceptualization, I.C., G.N., A.A. and W.L.; methodology, F.T., I.C., P.F., G.T.; formal analysis, W.L. and A.A.; data curation, F.T.; writing—original draft preparation, F.T., I.C., G.N., P.F., G.T.; writing—review and editing, A.A., W.L.; supervision, W.L.; funding acquisition, I.C., A.A. and W.L. All authors have read and agreed to the published version of the manuscript.

**Funding:** This research was supported through funds from the Region Latium (Italy) POR FESR 2014–2020 Key Enabling Technologies grant (project acronym: CRANIMA) to I.C. and W.L. and from the Università Cattolica del Sacro Cuore (Linea D.1 2017) to A.A. and W.L.

**Institutional Review Board Statement:** Not applicable.

**Informed Consent Statement:** Not applicable.

**Acknowledgments:** We would like to acknowledge the Regenerative Medicine Research Center (CROME) of Università Cattolica del Sacro Cuore.

**Conflicts of Interest:** The authors declare no conflict of interest.

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
