# Peer review of "Personalized Bone Reconstruction and Regeneration in the Treatment of Craniosynostosis"

_applsci, doi:10.3390/app11062649_

Round 1
Reviewer 1 Report
The presented paper is quite interesting. This review article provides precise and concise updates about craniosynostosis.
Although all I have some remarks to the Authors:
1) Lack of insertion/ reference to Table 1 and Table 2 in the main text.
2) Table 3: an advantage/disadvantage column of each available commercial fixation system would be very useful and will show complex information.
3) Is Figure 1b present the result of your study? It is no clear.
Author Response
The presented paper is quite interesting. This review article provides precise and concise updates about craniosynostosis.
Although all I have some remarks to the Authors:
1) Lack of insertion/ reference to Table 1 and Table 2 in the main text.
We thank the Referee for this observation. Changes have been implemented in the paragraphs 1.1 (see red text on page 2, lines 21-27) to mention the references that have been included in tables 1 and 2.
2) Table 3: an advantage/disadvantage column of each available commercial fixation system would be very useful and will show complex information.
We thank the Referee for this suggestion, which was addressed by introducing an additional column in Table 3, counting the rate of post-operative infections associated with the use of the selected rigid fixation systems, as a main and reproducible outcome measure comparable across different studies. See Table 3 on page 16, and the changes highlighted as red characters introduced in the text on page 15 line 33 to page 16 line 12.
3) Is Figure 1b present the result of your study? It is no clear.
Figure 1b shows the 3D reconstruction of a CT scan of consenting patients diagnosed with a metopic synostosis. Details were added in the figure legend.
Reviewer 2 Report
In this review, the authors carefully examined the treatment strategies in clinic as well as in animal research for syndromic and non-syndromic craniosynostosis, including surgical approaches, scaffold materials, biomolecules, and stem cell applications. The review is very informative and overall well-written, the exhaustive overview can provide very useful insight to the field. I support publication of this manuscript after the author addressing some minor concerns (see below).
- The authors should provide some degradation and mechanical information for the biomaterials listed (PCL, PLGA, etc.), which will be very useful for material selection purpose.
- The bioactive compounds and cell treatment part need to be strengthened and up-to-date. Quite a few molecules, signaling and stem cells have been tested to restore suture patency (periostin, Wnt signaling, Gli1+ stem cells). The author should include these up-to-date studies.
-
- J Transl Med. 2018 Apr 17;16(1):103.
- 2021 Jan 7;184(1):243-256.
- Instead of just listing all the studies that are available, the author should touch more on the pros and cons of these treatment strategies, such as surgical approaches, tissue engineering approaches, and stem cell therapies. What are the advantages of each treatment, and what are the challenging part?
Author Response
In this review, the authors carefully examined the treatment strategies in clinic as well as in animal research for syndromic and non-syndromic craniosynostosis, including surgical approaches, scaffold materials, biomolecules, and stem cell applications. The review is very informative and overall well-written, the exhaustive overview can provide very useful insight to the field. I support publication of this manuscript after the author addressing some minor concerns (see below).
- The authors should provide some degradation and mechanical information for the biomaterials listed (PCL, PLGA, etc.), which will be very useful for material selection purpose.
We thank the Referee for this suggestion, which we have taken into due account in the revision. This issue was in part already reported in the original version of the manuscript, but we have implemented additional details, accordingly. Find the implemented red text on page 18 lines 14-23.
- The bioactive compounds and cell treatment part need to be strengthened and up-to-date. Quite a few molecules, signaling and stem cells have been tested to restore suture patency (periostin, Wnt signaling, Gli1+ stem cells). The author should include these up-to-date studies.
We thank and agree with the Referee about this point. On this regard, additional details and updated data have been added in section 2.2 (see red text on page 18 line 50-to page 19 line 6), and in section 2.3 (see red text on page 21 – 24, and corresponding revisions in Table 5).
- Instead of just listing all the studies that are available, the author should touch more on the pros and cons of these treatment strategies, such as surgical approaches, tissue engineering approaches, and stem cell therapies. What are the advantages of each treatment, and what are the challenging part?
We have taken into due consideration this insightful observation in order to improve the overall quality of the manuscript. We have addressed this point by critically revising most sections of the paper (see red text throughout the manuscript body) and by introducing additional considerations in a broadly revised conclusive section (see section 4).
Reviewer 3 Report
The manuscript entitled, "Personalized bone reconstruction and regeneration in the treatment of craniosynostosis" is an interesting article. Before accepting this manuscript the following revisions are required.
Comments:
- Shear thinning and shape forming hydrogel are used for cranial defects. Few papers related to this should be incorporated in this review.
- FGF-18 & FGF-2 were extensively used growth factors in Craniofacial related area. Related this also to incorporated in this review.
- Natural bioactive phytochemicals also used cranial defects. Few examples should incorporated in this review.
Author Response
The manuscript entitled, "Personalized bone reconstruction and regeneration in the treatment of craniosynostosis" is an interesting article. Before accepting this manuscript the following revisions are required.
Comments:
- Shear thinning and shape forming hydrogel are used for cranial defects. Few papers related to this should be incorporated in this review.
We thank the referee for this suggestion. Data regarding the use of shear thinning hydrogels for cranial defects, with discussion of additional references, have been added in the current revised version of the manuscript, specifically in section 2.2 (see red text from page 19 line 44 to page 20 line 24, and corresponding revisions implemented in Table 4).
- FGF-18 & FGF-2 were extensively used growth factors in Craniofacial related area. Related this also to incorporated in this review.
We thank the referee for this precious observation. On this regard, additional details and updated data have been added in section 2.3 (see red text on page 21 lines 10-36).
- Natural bioactive phytochemicals also used cranial defects. Few examples should incorporated in this review.
We thank the referee for this suggestion. Accordingly, we have added some considerations and examples related to the employment of phytochemicals for cranial defects (see red text on page 24 lines 23-38).